# Sudden Sensorineural Hearing Loss after COVID-19 Vaccination: A Review of the Available Evidence through the Prism of Causality Assessment

**DOI:** 10.3390/vaccines12020181

**Published:** 2024-02-11

**Authors:** Hung Thai-Van, Haleh Bagheri, Marie-Blanche Valnet-Rabier

**Affiliations:** 1Department of Audiology and Otoneurological Evaluation, Hospices Civils de Lyon, 69003 Lyon, France; hung.thai-van@chu-lyon.fr; 2Institut Pasteur, Institut de l’Audition, 75015 Paris, France; 3Faculté de Médecine, Université Claude Bernard Lyon 1, 69100 Villeurbanne, France; 4Department of Medical and Clinical Pharmacology, Centre Régional de Pharmacovigilance de Toulouse, CIC1436, Hôpital Universitaire de Toulouse, 31000 Toulouse, France; haleh.bagheri@univ-tlse3.fr; 5Department of Clinical Pharmacology, Centre Régional de Pharmacovigilance et d’Information sur les Médicaments, Centre Hospitalier Universitaire de Besançon, 25000 Besançon, France

**Keywords:** mRNA COVID-19 vaccine, audiogram, case series study, disproportionality analysis, pharmacovigilance, positive rechallenge, postmarketing, spontaneous reporting, sudden sensorineural hearing loss, safety signal

## Abstract

Sudden sensorineural hearing loss (SSNHL), a rare audiological condition that accounts for 1% of all cases of sensorineural hearing loss, can cause permanent hearing damage. Soon after the launch of global COVID-19 vaccination campaigns, the World Health Organization released a signal detection about SSNHL cases following administration of various COVID-19 vaccines. Post-marketing studies have been conducted in different countries using either pharmacovigilance or medico-administrative databases to investigate SSNHL as a potential adverse effect of COVID-19 vaccines. Here, we examine the advantages and limitations of each type of post-marketing study available. While pharmacoepidemiological studies highlight the potential association between drug exposure and the event, pharmacovigilance approaches enable causality assessment. The latter objective can only be achieved if an expert evaluation is provided using internationally validated diagnostic criteria. For a rare adverse event such as SSNHL, case information and quantification of hearing loss are mandatory for assessing seriousness, severity, delay onset, differential diagnoses, corrective treatment, recovery, as well as functional sequelae. Appropriate methodology should be adopted depending on whether the target objective is to assess a global or individual risk.

## 1. Available Epidemiological Data on Sudden Sensorineural Hearing Loss

A rare audiological condition accounting for around 1% of all cases of sensorineural hearing loss [1], sudden sensorineural hearing loss (SSNHL) requires prompt diagnosis and treatment [2]. It is defined as the occurrence in less than 72 h of a sensorineural hearing loss of at least 30 dB over at least three successive audiometric frequencies [3]. According to the most recent epidemiological estimates, the historical incidence of SSNHL is 27 per 100,000 inhabitants per year in the USA: incidence was reported to increase with age, ranging from 11 per 100,000 in patients under 18 to 77 per 100,000 in patients aged 65 and over [4]. Most often unilateral, sudden onset deafness may be bilateral in the presence of a genetic or autoimmune context, in which case bilateral involvement may not be simultaneous [5]. All degrees of hearing loss can be observed in SSNHL as well as various patterns of audiometric loss, with different functional prognoses [6]. These may include audiometric loss in the low frequencies (possibly associated with vestibular disorders responsible for vertigo and balance disorders), in the mid- or in the high frequencies, plateau or notch audiometric loss, or even total deafness (cophosis) with the worst functional prognosis. In the event of sudden onset of hearing loss, the first step is to rule out retrocochlear pathology in the cerebellopontine angle, the most common of which is vestibular schwannoma [7,8]. An etiology can be identified in only one-third of cases (in decreasing order of frequency: infectious, otologic, traumatic, hematologic, neoplastic), and most SSNHL cases are idiopathic [9]. The pathophysiological mechanisms underlying the onset of idiopathic SSNHL have been described as viral [10,11], vascular [12,13], pressure-related [14,15], autoimmune [16,17], and genetic [18,19]. It has been reported that examining vestibular organ functions in subjects suffering from SSNHL could help identify its etiopathogenesis [20].

Among the otorhinolaryngological manifestations of SARS-CoV-2 infection, hearing loss is much less frequently reported than anosmia [21,22]. Nevertheless, several case studies have established a link between documented cases of SSNHL and the COVID-19 pandemic [23,24,25,26,27,28,29,30,31,32,33,34,35]. Virology research has shown a strong tropism of the SARS-CoV-2 spike glycoprotein for angiotensin-converting enzyme 2 receptors, notably in the vascular endothelium [36]. Post-COVID-19 SSNHL could therefore be due to vascular endotheliitis affecting the inner ear, cochlear nerve, or even central auditory pathway [37]. While a recent study pointed to differences in the occurrence of hearing disorders following COVID-19 infection, depending on the variant involved [38], this difference may not reach statistical significance [39]. Subjects with longer initial COVID-19 symptoms, however, may be more likely to suffer hearing loss after COVID-19 infection [39]. The World Health Organization (WHO) issued a safety signal in early 2022 concerning cases of SSNHL possibly linked to COVID-19 vaccination, triggering pharmacovigilance actions [40]. Sporadic cases of SSNHL were described following either adenoviral vector or mRNA-based vaccine administration after the first or second vaccine dose. Clinical presentations may correspond either to isolated deafness or to deafness combined with other otoneurological disorders, such as tinnitus or vertigo. In some cases, a history of otoneurological, cardiovascular or autoimmune disorders was documented [41,42,43,44].

Here, we sought to explain why, depending on the methodology used, contradictory results could be observed with regard to the incidence of SSNHL after COVID-19 vaccination. As a matter of fact, the frequency of an adverse event (AE) determines the methodological approaches to generating a safety signal. For very rare adverse events, pharmaco-epidemiological data could be less appropriate. Conversely, data from spontaneous notification and exhaustive pharmacovigilance monitoring may be more effective, particularly in the context of the COVID-19 vaccination campaign.

## 2. What Do We Currently Know about SSNHL Observed after mRNA COVID-19 Vaccination?

As shown in a systematic review by Liew et al. based on a 30 July 2023 search, post-marketing studies have been conducted to assess the incidence of SSNHL following COVID-19 vaccination in the following countries: United States, Finland, Israel, and France [45]. Using the Preferred Reporting Items for Systematic Review and Meta-Analysis guidelines, the authors identified nine studies dealing with COVID-19 vaccination and SSNHL, enabling them to estimate the range of incidence between 0.6 and 28 cases per 100,000 person-years. Of these nine studies, two, in fact, did not address SSNHL as a primary adverse event (AE) of COVID-19 vaccination, two were not conducted on a large scale, and only five were nationwide studies. Chronologically, the first national study to be published was performed in the United States from December 2020 to July 2021 [46], confirming preliminary results [47]. Using the Vaccine Adverse Events Reporting System (VAERS) system, the authors concluded that there was no difference between the historical incidence of SSNHL in the USA [4] and that observed after COVID-19 vaccination with tozinameran (Pfizer-BioNTech BNT162b2, Comirnaty™), elasomeran (Moderna mRNA-1273, Spikevax™), or ChAdOx1-S [recombinant] (AstraZeneca AZD1222 Vaxzevria™).

The same conclusions were drawn by a Finnish retrospective cohort study using data from the national healthcare register published a year later [48]. In the same timeframe, in 2022, another retrospective cohort study conducted by Clalit Health Services, Israel’s largest state-mandated health services organization, provided contrasting results [49]. The Israeli study highlighted the possibility of a higher risk of SSNHL following tozinameran administration. It should be noted that this risk, although documented, proved to be low (<1 per 100,000 vaccinated individuals). By the end of 2022, Chen et al. published an analysis of the VAERS-registered hearing disorders and identified an increased risk of hearing impairment after administration of both mRNA and virus vector COVID-19 vaccines compared to influenza vaccination in a real-life context [50]. Following the Finnish publication, in July 2023, Thai-Van et al. published data on all cases of SSNHL collected as part of the enhanced pharmacovigilance surveillance system set up in France [51]. The main difference with the aforementioned studies is that the French cases were all assessed and re-analysed by two otorhinolaryngologists specialized in otology and audiology, allowing for a clinical evaluation. The authors concluded that severe SSNHL may occur after mRNA COVID-19 vaccination, but only in rare situations. Subsequent to the systematic review by Liew et al. [45], a recent nationwide study conducted in Denmark by Damkier et al. found no significant difference between vaccinated and unvaccinated subjects with regard to the hospital diagnosis code of SSNHL [52]. However, in the Danish cohort, the prescription of corticosteroids by an otorhinolaryngologist, used as a proxy for out-of-hospital SSNHL diagnosis, was higher in the 21 days following mRNA COVID-19 vaccination.

In their systematic review, Liew et al. [45] commented on the lack of clinical data to support the reported incidence rates and highlighted, in particular, the absence of prognostic and functional recovery data in most studies. Regardless of the country in which the observational studies have been carried out, all authors emphasized both the low historical incidence of SSNHL and the low incidence of SSNHL after COVID-19 vaccination, as well as the need for relevant clinical data to fully explore a potential link between SSNHL and the COVID-19 vaccines. In addition to national studies conducted in the USA, Finland, Israel, France, and Denmark, we identified several case studies of SSNHL after COVID-19 vaccination [42,44,53,54,55,56,57]. Figure 1 shows the chronology of all relevant publications dealing with SSNHL after COVID-19 vaccination. 

## 3. Quantification of Sudden Sensorineural Hearing Loss after COVID-19 Vaccination

The clinical features and functional recovery of SSNHL depend to a large extent on the degree of hearing loss and the frequency bands involved [58]. Its prognosis varies according to whether the hearing loss is associated with other otoneurological disorders [59,60,61]. Therefore, the follow-up of any SSNHL requires the initial degree of hearing loss to be measured according to international standards and its clinical presentation to be documented as accurately as possible. According to the most common definition of pure-tone average found in epidemiological or population-based studies, the level of hearing impairment is calculated by averaging hearing thresholds at 500, 1000, 2000, and 4000 Hz (https://www.nidcd.nih.gov/health/statistics/what-numbers-mean-epidemiological-perspective-hearing (accessed on 29 December 2023)). In the absence of reference audiometry, the diagnosis of SSNHL can be made by comparing the hearing thresholds of the affected ear with those of the contralateral ear. The fact that the post-marketing observational studies conducted in the USA [46], Finland [48], Israel [49], and Denmark [52] may have been based on diagnosis codes and not on audiological measurements as such is a first factor explaining the lack of homogeneity of their conclusions. Nieminen et al. [48] stressed that more conclusive studies should meet the international diagnosis criteria for SSNHL, rather than just rely on diagnosis codes from the care register for healthcare. 

In line with recent research on SSNHL [62,63], affected subjects must be classified according to a classification system derived from Siegel’s criteria [64]. This classification enables precise assessment of hearing impairment, intending to quantify SSNHL and then potential hearing recovery based on standardized objective criteria (see Table 1). Such quantification comprises five grades covering all possible degrees of hearing loss: namely, slight (grade 1), mild (grade 2), moderate to moderately severe (grade 3), severe (grade 4), and profound (grade 5). 

## 4. Evaluation of Hearing Recovery in SSNHL Post-COVID-19 Vaccination: Benefits of Active Audiogram-Based Surveillance

Following the publication of the large-scale observational studies carried out in the USA [46] and Israel [49] with contradictory results, it was pointed out that neither study had attempted to characterize the duration of post-vaccination SSNHL, nor the concomitant presence of potential risk factors [65]. Along the same lines, the reliability of passive reporting and retrospective comparisons of administrative data for both diagnosis and surveillance of SSNHL has been described as subject to several biases [66]. Given that VAERS is, first and foremost, a national system for early detection of vaccines’ AEs, it is to be feared that the reports were written incompletely or imprecisely or were simply the result of a coincidence. Such an alert system allows anyone in the USA to report a potential AE after receiving a vaccine. In any case, these reports are difficult to verify and may underestimate the actual incidence of AEs or, on the contrary, be redundant. In the study by Formeister et al. [46], cases of self-reported hearing loss were not audiologically documented, nor was their time of onset in relation to the vaccine injection. There were, in addition, no data available on hearing recovery outcomes.

As with any SSNHL, recovery following COVID-19 vaccine administration may be complete, partial, slight, or absent, or it may result in a non-serviceable ear. The levels of hearing recovery can be rated using a classification derived from Siegel’s criteria [62,63,64]. In this classification, the distinction between the “no improvement status“ and the “non-serviceable ear status“ allows for the fact that only patients with serviceable hearing levels can benefit from conventional hearing amplification (i.e., hearing aids), while patients with non-serviceable ears are generally referred for surgical cochlear implantation. 

The French nationwide PV study carried out under the aegis of the public health authorities estimated the rate of SSNHL reporting after COVID-19 mRNA vaccination per one million doses, using a cut-off value of 21 days post-exposure [51]. This 21-day post-vaccination limit has been approved by the Safety Platform for Emergency vACcines (SPEAC) and the Brighton Collaboration’s guidelines for Sensorineural Hearing Loss [67]. For each case of SSNHL reported, a request was sent to the regional PV centre involved to access the patient’s medical file, audiogram, and neuroradiological examinations. The authors retained only cases of SSNHL medically documented by audiological measurements in the absence of any concomitant etiology of hearing loss, such as retrocochlear disorder (assessed by systematic MRI examination), endolymphatic hydrops or head injury and concussion. They also investigated the potential facilitative role of risk factors, whether autoimmune, cardiovascular or otoneurological. In one-third to one-half of identified SSNHL cases, other audio-vestibular symptoms such as tinnitus or balance disorders were also present. Autoimmune, cardiovascular or otoneurological risk factors were present in approximately 30% of identified cases. Steroids were administered orally in half of SSNHL cases. For every case identified, a minimum follow-up of 3 months made it possible to assess the level of hearing recovery, if any. Hearing recovery outcomes in the French study were in line with those observed in historic SSNHL outside the vaccination context [61] as shown in Table 2. Of note, cases of positive rechallenge have been documented for both tozinameran and elasomeran. The cases of SSNHL were found to represent, respectively, 0.63% and 0.76% of the total serious adverse reactions recorded for tozinameran and elasomeran in France.

## 5. When an Adverse Event Becomes an Adverse Effect and Then a Signal

The Causality assessment (CA) in PV is a process well described and based on different methodologies [68,69,70]. For vaccines, in the 2019 update of the CA of an adverse event following immunization (AEFI), WHO provided a user manual for persons implicated in immunization programmes [71]. With this user guide, WHO developed an AEFI CA support tool available online to help people implicated in vaccine PV systems. Thus, several definitions regarding vaccination are available. An AEFI is any untoward medical occurrence which follows immunization and which does not necessarily have a causal relationship with the usage of the vaccine. The AE may be any unfavourable or unintended sign, abnormal laboratory finding, symptom or disease. A causal association is defined as “a cause-and-effect relationship between a causative factor and a disease with no other factors intervening in the process”. Among the different given definitions, WHO made a difference between five cause-specific definitions, i.e., vaccine product-related reaction, vaccine quality defect-related reaction, immunization error-related reaction, immunization anxiety-related reaction/immunization stress-related response, and coincidental event. Hence, the regulators introduced the notion of causality, and the term event became a reaction or effect [72].

The CA, from the WHO point of view, is “the systematic review of data about an AEFI case; it aims to determine the likelihood of a causal association between the event and the vaccine received” and “the quality of the CA depends upon (1) the performance of the AEFI reporting system in terms of responsiveness, effectiveness and quality of investigation and reports, (2) the availability of adequate medical and laboratory services and access to background information; and (3) the quality of the causality review process”.

Based on Bradford Hill criteria [73], several criteria are relevant to establishing causality at the population level: temporal relationship, the strength of the association, dose-response relationship, consistency of evidence, specificity, and biological plausibility and coherence. The question to answer is ”Can the given vaccine cause a particular adverse event?”

At the individual level, the question to answer is more likely to be “Did the vaccine given to a particular individual cause the particular event reported?” In this situation, the scientific basis for the criteria which are assessed in the process includes temporal relationship, the definitive proof that the vaccine caused the event, biological plausibility, consideration of alternative explanations, and prior evidence that the vaccine in question could cause a similar event in the vaccine, notably with the concept of rechallenge.

Only once the event is considered as a reaction can a signal evaluation be opened with the objective of drawing conclusions on the presence or absence of a causal association between an AE and a vaccine and identifying a need for additional data collection or considering risk minimisation measures.

It is essential to recognise that the CA of an AEFI in an individual patient is an exercise in medical differential diagnosis as it could be performed to diagnose diabetes or multiple sclerosis.

With the beginning of the COVID-19 vaccination campaign as soon as January 2021, The French National Agency for the Safety of Medicines and Health Products (ANSM) ordered national active surveillance for each new COVID-19 vaccine. Several regional centres of PV were implicated as experts, and the network of the 30 regional PV centres were deeply invested in the analysis and the medical documentation of spontaneous cases with a priority on the serious cases. The French PV network is organised around 30 regional PV centres, all located in a university hospital centre to be near patients and health professionals. This specific organisation allows short information circuits and rapid communications, as shown in Figure 2 [74,75,76].

The first goal of PV is the identification of new adverse drug reactions (ADR) from all spontaneous notifications received by the regional PV centre. Through pharmacological expertise, health professionals from PV could routinely identify safety signals, i.e., situations that could lead to a public health problem and for which the authorities must take measures to reduce the risk for patients. As stated by Jonville-Bera et al., “the role of the pharmacovigilant is rather to detect unexpected or rare ADR, which escaped the scrutiny of a large randomized clinical trial” [74]. To reach this ultimate goal, pharmacovigilants, as experts in drug adverse reactions and iatrogenic disease, are always working closely with specialists and clinicians to obtain all the details of a patient’s medical history. 

For example, from the COVID-19 vaccination campaign, myocarditis, defined as an adverse event, was rapidly validated as a safety signal and recognised as an adverse effect of the mRNA COVID-19 vaccine. In France, as soon as April 2021, and in a parallel way to the safety signal from Israel, we received five unexpected myocarditis events in young patients [77,78]. Thus, ANSM validated a safety signal at the national level, and concomitantly, the European Medicine Agency (EMA) and the PV Risk Assessment Committee (PRAC) opened a signal procedure for evaluating the mRNA vaccine role in this event.

During the following months, regional PV centres received other myocarditis cases in the context of vaccination campaigns implemented in the young population. To quantify the potential risk and qualify myocarditis as an adverse effect, pharmacovigilants conducted a rigorous analysis based on clinical symptoms and the course of the disease and collected all diagnostic criteria according to the Brighton recommendation [66,79], allowing the classification of cases as suspected, probable, or confirmed. A reporting rate was calculated between the mRNA vaccine among people under 30 years old and the number of injected doses. The difference found was presented to ANSM and confirmed with a case-control analysis using administrative data [80]. This safety signal was further assessed and confirmed through several pharmacoepidemiology studies. Data about myocarditis are still under publication, whereas the signal is validated by health authorities and validated as an adverse effect [77,81]. Contrary to SSNHL, myocarditis epidemiology is better described, also in the context of COVID-19 infection. 

In the context of the COVID-19 vaccination campaign in Europe, the reinforced surveillance system in place relies on the complementary approaches of PV and pharmacoepidemiology [82].

The process of safety signal validation could be more or less rapid according to collected data, the frequency of the suspected effect, and the consequences. Dhodapakar et al. assessed the role and impact of safety signals from the US Food and Drug Administration’s Adverse Event Reporting System (FAERS) database on subsequent regulatory actions by the administration [83]. Spontaneous reporting systems and analyses of aggregated cases in databases such as FAERS are a cornerstone in generating post-marketing safety signals. For a subset of 82 potential safety signals, a literature search identified 1712 relevant publications corresponding to case reports or case series for 70% of them [84].

## 6. Two Different but Complementary Approaches to Safety Signal Evaluation

Due to their design, objectives, and number of subjects included, clinical trials are insufficient to assess the safety of new drugs, particularly for rare events. Post-marketing pharmacovigilance datas, however used, are essential for monitoring the safety of marketed medicines, and the global landscape of pharmacovigilance research is expanding [85]. However, few data are available on methods leading to drug withdrawal decisions. Lasser et al. have shown that 10.2% of the 548 new drugs approved in the USA between 1975 and 1999 were the subject of a new black box warning or were withdrawn. Half of these withdrawals took place within two years post-marketing [86]. In France, a total of 21 drugs were withdrawn from the market for safety reasons between 1998 and 2004. For 12 of them, the scientific evidence leading to withdrawal came from spontaneous case reports or case series [87]. In Spain, 22 drugs were withdrawn between 1990 and 1999. In 82% of cases, the evidence supporting the drug withdrawal came from individual case reports, case series or a combination of data from randomized clinical trials and case reports [88]. In addition to pharmacovigilance data from daily practice, pharmacoepidemiological studies complete the approach to signal detection, such as disproportionality analyses [89,90,91,92].

In the French study, all SSNHL cases after mRNA COVID-19 vaccination were analysed by both pharmacovigilant and otolaryngologist, enabling the selection of only cases documented based on an audiogram or medical report, chronological data, and any additional data [50]. The authors were thus able to estimate the minimum rate of this adverse reaction based on “robust” cases, and we identified a few cases with a new positive rechallenge, suggesting a high probability of a causal relationship [93]. Under-reporting remains one of the main limitations of the spontaneous reporting method. The authors cannot exclude this limitation, which may have biased the estimated rate of SSNHL. However, following the introduction of mRNA COVID-19 vaccines in France, the strengthened pharmacovigilance system made considerable efforts to publicize the mandatory reporting of any suspected adverse reactions following the administration of these vaccines. In this proactive post-marketing surveillance context, the under-reporting rate should, therefore, be lower than in a routine pharmacovigilance process [94]. In France, for example, more than 190,000 cases of adverse reactions were reported to the pharmacovigilance system. Of these, cases of SSNHL accounted for 0.63% and 0.76%, respectively, of the total serious adverse reactions recorded for tozinameran and elasomeran (https://ansm.sante.fr/actualites/point-de-situation-sur-la-surveillance-des-vaccins-contre-le-covid-19-periode-du-14-04-2023-au-8-06-2023 (accessed on 29 December 2023)). Another limitation of the French study is the absence of a control group. However, as mentioned above, the role of PV is to identify specific unexpected adverse effects that were not identified before marketing, to validate the cause-and-effect relationship and to generate a signal. Comparison with a control group could confirm the signal generated and quantify the risk ratio, and/or identify the risk in a particular population. Following their first results, the authors performed a disproportionality analysis using the French pharmacovigilance database and found that the reporting odds ratio for hearing loss was 1.94, CI 95% [1.40–2.77] for mRNA COVID-19 vaccines versus other vaccines (submitted). This signal has been, in addition, significant since April 2021 [95].

The prospect of a world without pharmacovigilance or pharmacoepidemiology has been imagined here only to underline the importance of either approach to monitoring drug safety. However, it is essential to highlight the risks of either situation. Currently, no signal detection activity from pharmacoepidemiological studies can rival the results obtained by PV methods such as spontaneous notifications [96]. PV remains unrivalled in detecting the risk of adverse events whose frequency is extremely low in the population outside drug use, as it is precisely the case for SSNHL. In such situations, medico-administrative databases, however large, may be insufficient to reveal the risk of rare adverse reactions. Using all cumulative data, signal detection in spontaneous reporting systems achieved higher specificity and sensitivity than administrative data [97]. The rarity of SSNHL cases after mRNA CVOID-19 vaccines supports the relevance of spontaneous reporting, which is useful for identifying rare and idiosyncratic safety issues, thus enabling signal generation. This signal was also supported by pathophysiological, pharmacological, meta-analytical, and/or pharmacoepidemiological data with disproportionality analysis [36,98,99].

## 7. Rare Vaccine Adverse Events: Challenges and Perspectives

Post-marketing surveillance of drugs is regulated and based on a methodology well described at the expert level. Based on the reporting of real-life data and depending on a voluntary process, any PV system has the same limitations, such as under-reporting and lack of data quality. Every actor in the PV process should be involved in collecting the most exhaustive data. Pharmacoepidemiological studies using PV databases or administrative data face the same completeness and data quality limitations. The foundations of an efficient PV system lie in collaboration between pharmacologists and clinicians to validate spontaneous reports using relevant criteria. Patient participation in the PV system is also essential to reduce under-reporting. In this analysis of all available data published, we discussed the benefits and limitations of currently available studies on SSNHL occurring after COVID-19 mRNA vaccination and the difficulty of establishing a causal relationship between vaccination and SSNHL. We believe the added value of PV studies to explore such a rare event cannot be questioned. To this end, a comprehensive approach would require both high-quality data and a precise medical assessment of each notified case to carry out disproportionality analyses in relation to PV best practice. Close collaboration between a strengthened national PV network and clinical experts would also help avoid, as far as possible, under- or over-estimating the true incidence of post-vaccination SSNHL.

## Figures and Tables

**Figure 1 vaccines-12-00181-f001:**
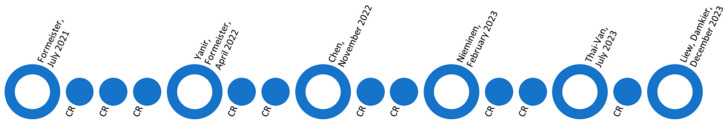
Timeline of the leading publications dealing with SSNHL post-COVID-19 vaccination. A comprehensive search of PubMed (pubmed.ncbi.nlm.nih.gov, accessed on 1 February 2024) and Embase (embase.com, accessed on 1 February 2024) was conducted using the keywords “COVID-19 vaccination” and “hearing loss”. CR: Case Report.

**Figure 2 vaccines-12-00181-f002:**
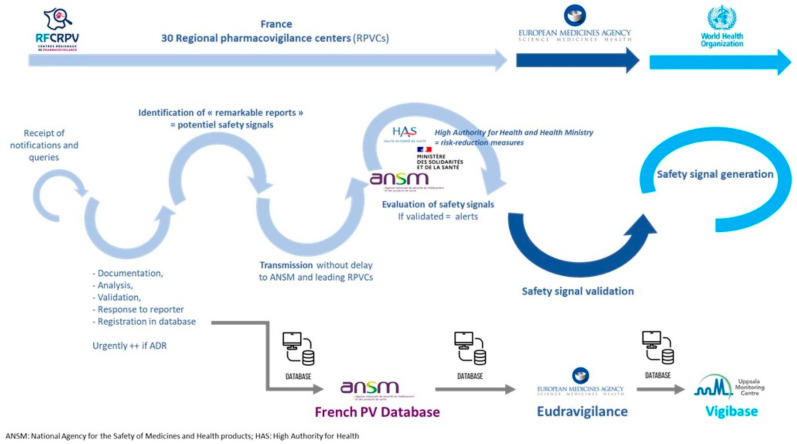
Signal detection in pharmacovigilance: example of the French system.

**Table 1 vaccines-12-00181-t001:** Modified Siegel’s criteria for grades of hearing impairment.

Siegel’s Grade of Hearing Loss	Hearing Loss Degree (dB HL)
Grade 1	≤25
Grade 2	26–40
Grade 3	41–70
Grade 4	71–90
Grade 5	>90

**Table 2 vaccines-12-00181-t002:** Hearing recovery outcomes by grade of hearing impairment in the French study (adapted from Thai-Van et al. 2023 [51]).

	Number of SSNHL Cases after mRNA COVID-19 Vaccination = 171 (From January 2021 to February 2022, in France)	
Hearing Outcome Hearing loss N (%)	Complete Recovery	Partial Recovery	Slight Improvement	No Improvement	Non-Serviceable Ear	Total Number of Cases/Hearing Loss Grade
**Grade 1**	16 (10)	0	0	15 (9)	0	31 (18)
**Grade 2**	11 (6.5)	6 (3)	0	25 (15)	0	42 (24.5)
**Grade 3**	11 (6.5)	8 (5)	1 (0.5)	33 (19.5)	0	53 (31)
**Grade 4**	1 (0.5)	4 (2)	3 (1.5)	14 (8)	0	22 (13)
**Grade 5**	0	2 (1)	4 (2)	0	17 (10)	23 (13.5)
**Total number of cases/outcome**	39 (23)	20 (12)	8 (4)	87 (51)	17 (10)

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
