# Peer review of "Sudden Sensorineural Hearing Loss after COVID-19 Vaccination: A Review of the Available Evidence through the Prism of Causality Assessment"

_vaccines, 2024, doi:10.3390/vaccines12020181_

Round 1
Reviewer 1 Report
Comments and Suggestions for Authors
REVIEW - How Difficult is it to Assess Sudden Sensorineural Hearing Loss as a Potential Adverse Effect of COVID-19 Vaccines? A Critical Review of the Most Relevant Pragmatic Approaches
This study is a critical review of the approaches used in other research studies investigating the potential adverse effects of COVID-19 vaccines on sudden sensorineural hearing loss (SSNHL). The study highlights the importance of audiogram-based surveillance in assessing hearing recovery and functional outcomes in SSNHL cases. It discusses the limitations of observational studies based on self-reported hearing loss and emphasizes the need for collaboration between pharmacovigilance experts and audiology specialists. The study also mentions the rarity of SSNHL cases following COVID-19 vaccination, which supports the relevance of spontaneous reporting in signal detection. However, the study acknowledges the limitations of under-reporting and data quality in spontaneous reporting systems and highlights the importance of active collaboration between healthcare professionals and patients to minimize under-reporting. In conclusion, the study provides a critical evaluation of the existing research on SSNHL after COVID-19 vaccination and emphasizes the need for comprehensive and collaborative approaches in pharmacovigilance research.
Corrections:
1. It is defined as the occurrence in less than 72 hours of a sensorineural hearing loss of at least 30 dB over at least 3 successive audiometric frequencies occurring within <72 hours. (page 1, row 37) – double mentioning of 72 hours
2. wiith (page2, row 90) – spelling
3. ae (page 5, row 211) – small caps
4. challenge (page 7, row 288) – rechallenge?
Print with minor changes.
Comments on the Quality of English LanguageCorrections:
1. It is defined as the occurrence in less than 72 hours of a sensorineural hearing loss of at least 30 dB over at least 3 successive audiometric frequencies occurring within <72 hours. (page 1, row 37) – double mentioning of 72 hours
2. wiith (page2, row 90) – spelling
3. ae (page 5, row 211) – small caps
4. challenge (page 7, row 288) – rechallenge?
Print with minor changes.
Author Response
Dear reviewer,
Thank you very much for your very thorough and detailed reading of our manuscript.
Yours faithfully,
- It is defined as the occurrence in less than 72 hours of a sensorineural hearing loss of at least 30 dB over at least 3 successive audiometric frequencies occurring within <72 hours. (page 1, row 37) – double mentioning of 72 hours
Thank you, this is now corrected in the revised manuscript.
- wiith (page2, row 90) – spelling
Thank you for your attention. The spelling mistake has been corrected in the revised manuscript.
- ae (page 5, row 211) – small caps
This has been modified accordingly.
- challenge (page 7, row 288) – rechallenge?
Thank you. This correction has been made.
Reviewer 2 Report
Comments and Suggestions for Authors
Thank you for the opportunity to review the manuscript entitled “How Difficult is it to Assess Sudden Sensorineural Hearing Loss as a Potential Adverse Effect of COVID-19 Vaccines? A Critical Review of the Most Relevant Pragmatic Approaches” , which was aimed to “sought to explain the conflicting results according to the method of the study used with regard to the post-marketing surveillance outcomes of SSNHL after COVID-19 vaccination… also aimed to define pragmatic approaches for active surveillance of COVID-19 vaccines adverse events (AEs), from the stage of spontaneous reporting to the generation of a safety signal if warranted… investigated the extent to which the rarity, or non-rarity, of the adverse event under surveillance may impact on the appropriateness of different pharmacovigilance and pharmaco-epidemiology approaches in the context of COVID-19 vaccination campaigns.”
The topic is interesting, and it is of public concern. However, the title, the abstract and the main body of the document are not congruent with each other. Even further, the title is misleading, according to the broad aims of the study.
- The majority of the content related to hearing loss overlaps with a previous report by the same first author.
- There is no evidence of an objective and comprehensive review of the studies available.
- The abstract includes information from a study performed in a single country in 2007, which is not directly related to this report; while the information provided at the beginning of the Introduction is from year 1996.
- The description of the epidemiology studies is incomplete.
- The structure and sequence of the manuscript is confusing.
- The conclusions are derived from international guidelines, but not from this report.
Comments on the Quality of English LanguagePunctuation errors require review.
Author Response
Dear reviewer,
Thank you very much for your constructive and helpful comments. Please be assured that we have done our best to clarify each point you raised. Below you will find our point-by-point responses to each of them. In particular, the title of the manuscript has been changed into “Sudden Sensorineural Hearing Loss after COVID-19 Vaccination: A Review of the Available Evidence through the Prism of Causality Assessment “, its sections have been renamed, and both the abstract and the body of document have been fully revised to meet your requirements.
Yours faithfully,
The authors
Thank you for your comment, which we have taken fully into account as follows:
1) The section specifically devoted to the critical review of available studies has been rewritten, expanded and updated. It is now entitled “What do we currently know about SSNHL observed after mRNA COVID-19 vaccination? “ (see page 2 lines 84 to 142 in the revised manuscript). In addition, a new figure has been added to illustrate the results of our systematic review (Fig. 1 in the revised manuscript).
2) The section “Evaluation of Hearing Recovery in SSNHL post-COVID 19 Vaccination: Benefits of Active Audiogram-Based Surveillance” has been amended to emphasize the value of a clinically based pharmacovigilance approach to assess the causality at the individual level (see page 5 lines 201 to 204 in the revised manuscript). Please note that the table "Modified Siegel's criteria for hearing recovery outcomes" included in the first submission has been removed from the revised manuscript, as it has already been published elsewhere (Thai-Van et al. 2023).
- There is no evidence of an objective and comprehensive review of the studies available.
We would like to thank you for this comment. We have taken it fully into account and carried out a comprehensive search of PubMed (pubmed.ncbi.nlm.nih.gov, accessed on February 1, 2024) and Embase (embase.com, accessed on February 1, 2024), using the keywords "COVID-19 vaccination" and "hearing loss". In doing so, we were able to complete the previous literature review, which included only studies published up to July 30, 2023 (Liew et al., 2023). We were thus able to add a Danish national study (Damkier, P.; Cleary, B. ; Hallas, J.; Schmidt J.H.; Ladebo, L.; Jensen, P.B.; et al. Sudden Sensorineural Hearing Loss Following immunization with BNT162b2 or mRNA-1273: A danish population-based cohort study. Otolaryngol Head Neck Surg. 2023, 169:1472-1480). In addition, we were able to identify the following case studies that were not included in the previous systematic review by Liew et al. (2023):
- Cohen, M.O. ; Tamir, S.O. ; O' Rourke, N. ; Marom, T. Audiometry-confirmed Sudden Sensorineural Hearing Loss incidence among COVID-19 patients and BNT162b2 vaccine recipiens. Otol Neurotol. 2023, 44, e68-e72. doi: 10.1097/MAO.
- Fisher, R. ; Tarnovsky, Y. ; Hirshoren, N. ; Kaufman, M. ; Stern, S.S. The association between COVID-19 vaccination and idiopathic Sudden Sensorineural Hearing Loss, clinical manifestation and outcomes. Eur Arch Otorhinolaryngol. 2023, 280, 3609-3613. doi: 10.1007/s00405-023-07869-2.
- Pisani, D.; Leopardi, G.; Viola. ; Scarpa, A.; Ricciardiello, F.; Cerchiai, N.; et al. Sudden sensorineural hearing loss after covid-19 vaccine; A possible adverse reaction? Otolaryngol Case Rep. 2021, 21:100384. doi: 10.1016/j.xocr.2021.100384.
- Zoccali, F.; Cambria, F.; Colizza, A.; Ralli, M.; Greco, A.; de Vincentiis, M.; et al Sudden Sensorineural Hearing Loss after third booster of COVID-19 vaccine administration. Diagnostics (Basel). 2022, 12, 2039. doi: 10.3390/diagnostics12092039.
- Andrade, J. ; Sessa, L. ; Ephrat, M. ; Truong, J. ; Di Gregorio, R. A Case Report of Sudden Sensorineural Hearing Loss (SSNHL) after administration of the COVID-19 Vaccine. J Pharm Pract. 2022, 19, 8971900221147584. doi: 10.1177/08971900221147584.
- Ekobena, P.; Rothuizen, L.E.; Bedussi, F.; Guilcher, P.; Meylan, S.; Ceschi, A.; et al. Four cases of audio-vestibular disorders related to immunisation with SARS-CoV-2 mRNA vaccines. Int J Audiol . 2022, 5,1-5. doi: 1080/14992027.2022.2056721
- Jeong, J. ; Choi, H.S. A Sudden Sensorineural Hearing Loss after COVID-19 vaccine.Int J Infect Dis. 2021, 113 : 341-343. doi: 10.1016/j.ijid.2021.10.025.
A new figure (Figure 1) has been added in the revised manuscript and now shows all the national post-marketing studies carried out in chronological order in the USA, Israel, Finland, France and Denmark, as well as all the case studies available as of February 1, 2024.
- The abstract includes information from a study performed in a single country in 2007, which is not directly related to this report; while the information provided at the beginning of the Introduction is from year 1996.
Thank you. We have modified the abstract to make it consistent with the beginning of the revised manuscript. In the first section, data on the incidence of SSNHL, a rare and serious condition, are the most recent to have been published. This incidence is consistent with that often reported in high-income countries, ranging from 5 to 20 per 100,000 person-years (see Nieminen et al., 2023).
- The description of the epidemiology studies is incomplete.
The revised manuscript includes in the section 2 entitled " What do we currently know about SSNHL observed after mRNA COVID-19 vaccination?” a completed description of all the national post-marketing studies carried out, focusing on the methodology employed in each country, the results obtained and the advantages and disadvantages of each. For example, we now explain how the Danish national study (Damkier et al. 2023), although based on the diagnostic codes used to track inpatient SNNHL cases, also used otorhinolaryngologists' prescription of corticosteroids within 21 days of COVID-19 vaccination as a proxi for out-of-hospital diagnosis of SSNHL (see page 2 lines 84 to 142 in the revised manuscript).
- The structure and sequence of the manuscript is confusing.
For ease of reading, the revised manuscript is now structured into seven sections as follows:
- Available epidemiological data on sudden sensorineural hearing loss
- What do we currently know about SSNHL observed after mRNA COVID-19 vaccination?
- Quantification of Sudden Sensorineural Hearing Loss after COVID-19 Vaccination
- Evaluation of Hearing Recovery in SSNHL post-COVID 19 Vaccination: Benefits of Active Audiogram-Based Surveillance
- When an adverse event becomes an adverse effect and then a signal
- Two different but complementary approaches to safety signal evaluation
- Rare vaccine adverse events: challenges and perspectives
Please see the tracked changes we have made in each section.
- The conclusions are derived from international guidelines, but not from this report.
The final section, now entitled “Rare vaccine adverse events: challenges and perspectives”, aims to highlight the complementary nature of pharmaco-epidemiological and pharmacovigilance studies when assessing an adverse event as rare as SSNHL following COVID-19 vaccination. While pharmaco-epidemiological studies provide a quantitative assessment, pharmacovigilance studies provide a qualitative assessment focusing on causality.
Thank you.
Reviewer 3 Report
Comments and Suggestions for Authors
The authors wrote a literary review about Sudden Sensorineural Hearing Loss as a Potential Adverse Effect of COVID-19 Vaccines! The aim of the authors is very important, to investigated the extent to which the rarity, or non-rarity, of the adverse event under surveillance may impact on the appropriateness of different pharmacovigilance and pharmaco-epidemiology approaches in the context of COVID-19 vaccination campaigns.
The article is a narrative review and not a systematical review. In any case I think that PRISMA criteria should be adopted to selected correctly the manuscripts, according to MDPI guidelines.
I think that in the discussion you should talk about the treatment used for SSNHL! What about the diagnosis? How many patients performed MRI after SSNHL?
Regarding the SSNHL and the use of vaccination, I think that the use of VEMPs should help us to define the cause (vascolar, virus-protein like). Please use this reference: Ciodaro F, Freni F, Alberti G, Forelli M, Gazia F, Bruno R, Sherdell EP, Galletti B, Galletti F. Application of Cervical Vestibular-Evoked Myogenic Potentials in Adults with Moderate to Profound Sensorineural Hearing Loss: A Preliminary Study. Int Arch Otorhinolaryngol. 2020 Jan;24(1):e5-e10.
Author Response
Dear reviewer,
Thank you very much for your constructive and helpful comments. Please be assured that we have done our best to clarify each point you raised. Below you will find our point-by-point responses to each of them.
Yours faithfully,
The authors
The article is a narrative review and not a systematical review. In any case I think that PRISMA criteria should be adopted to selected correctly the manuscripts, according to MDPI guidelines.
We would like to thank you for this comment. We have taken it fully into account and carried out a comprehensive search of PubMed (pubmed.ncbi.nlm.nih.gov, accessed on February 1, 2024) and Embase (embase.com, accessed on February 1, 2024), using the keywords "COVID-19 vaccination" and "hearing loss". In doing so, we were able to complete the previous literature review, which included only studies published up to July 30, 2023 (Liew et al., 2023). We were thus able to add a Danish national study (Damkier, P. ; Cleary, B. ; Hallas, J. ; Schmidt J.H. ; Ladebo, L. ; Jensen, P.B. ; et al. Sudden Sensorineural Hearing Loss Following immunization with BNT162b2 or mRNA-1273: A danish population-based cohort study. Otolaryngol Head Neck Surg. 2023, 169:1472-1480). Further, we were able to identify the following case studies that were not included in the previous systematic review:
- Cohen, M.O. ; Tamir, S.O. ; O' Rourke, N. ; Marom, T. Audiometry-confirmed Sudden Sensorineural Hearing Loss incidence among COVID-19 patients and BNT162b2 vaccine recipiens. Otol Neurotol. 2023, 44, e68-e72. doi: 10.1097/MAO.
- Fisher, R. ; Tarnovsky, Y. ; Hirshoren, N. ; Kaufman, M. ; Stern, S.S. The association between COVID-19 vaccination and idiopathic Sudden Sensorineural Hearing Loss, clinical manifestation and outcomes. Eur Arch Otorhinolaryngol. 2023, 280, 3609-3613. doi: 10.1007/s00405-023-07869-2.
- Pisani, D.; Leopardi, G.; Viola. P.; Scarpa, A.; Ricciardiello, F.; Cerchiai, N.; et al. Sudden sensorineural hearing loss after covid-19 vaccine; A possible adverse reaction? Otolaryngol Case Rep. 2021, 21:100384. doi: 10.1016/j.xocr.2021.100384.
- Zoccali, F.; Cambria, F.; Colizza, A.; Ralli, M.; Greco, A.; de Vincentiis, M.; et al Sudden Sensorineural Hearing Loss after third booster of COVID-19 vaccine administration. Diagnostics (Basel). 2022, 12, 2039. doi: 10.3390/diagnostics12092039.
- Andrade, J. ; Sessa, L. ; Ephrat, M. ; Truong, J. ; Di Gregorio, R. A Case Report of Sudden Sensorineural Hearing Loss (SSNHL) after administration of the COVID-19 Vaccine. J Pharm Pract. 2022, 19, 8971900221147584. doi: 10.1177/08971900221147584.
- Ekobena, P.; Rothuizen, L.E.; Bedussi, F.; Guilcher, P.; Meylan, S.; Ceschi, A.; et al. Four cases of audio-vestibular disorders related to immunisation with SARS-CoV-2 mRNA vaccines. Int J Audiol . 2022, 5,1-5. doi: 1080/14992027.2022.2056721
- Jeong, J. ; Choi, H.S. A Sudden Sensorineural Hearing Loss after COVID-19 vaccine.Int J Infect Dis. 2021, 113 : 341-343. doi: 10.1016/j.ijid.2021.10.025.
A new figure (Figure 1) has been added in the revised manuscript and now shows all the national post-marketing studies carried out in chronological order in the USA, Israel, Finland, France and Denmark, as well as all the case studies available as of February 1, 2024.
I think that in the discussion you should talk about the treatment used for SSNHL! What about the diagnosis? How many patients performed MRI after SSNHL?
In the only national post-marketing study using clinical and audiological data (Thai-Van et al., JMIR Public Health 2023), only medically-documented cases of SSNHL were considered: in particular, audiometric measurements were performed in all subjects, and patients were all screened for a tumoral aetiology through an MRI of the posterior fossa. Among a total of 400 extracted cases, a tumoral aetiology was found in 6 of them. Steroids were administered orally in 67 (47%) Tozinameran cases and in 16 (55%) Elasomeran cases (for further detail, see Thai-Van et al., JMIR Public Health 2023). This is now clearly stated in the 4th section of the revised manuscript, entitled “Evaluation of Hearing Recovery in SSNHL post-COVID 19 Vaccination: Benefits of Active Audiogram-Based Surveillance”. (see page 5 lines 201 to 204 and 209 to 210 in the revised manuscript).
Regarding the SSNHL and the use of vaccination, I think that the use of VEMPs should help us to define the cause (vascolar, virus-protein like). Please use this reference: Ciodaro F, Freni F, Alberti G, Forelli M, Gazia F, Bruno R, Sherdell EP, Galletti B, Galletti F. Application of Cervical Vestibular-Evoked Myogenic Potentials in Adults with Moderate to Profound Sensorineural Hearing Loss: A Preliminary Study. Int Arch Otorhinolaryngol. 2020 Jan;24(1):e5-e10.
Thank you for this very pertinent comment. Accordingly, the usefulness of vestibular evoked myogenic potentials (VEMPs) in identifying the etiopathogenesis of SSNHL is now acknowledged in the introduction section with the relevant reference (see page 2 lines 54 to 56 in the revised manuscript).
Round 2
Reviewer 2 Report
Comments and Suggestions for Authors
Please supplement reference one with an updated reference.